# Physical Activity and Screen Time among Hungarian High School Students during the COVID-19 Pandemic Caused Distance Education Period

**DOI:** 10.3390/ijerph182413024

**Published:** 2021-12-10

**Authors:** Zsolt Bálint Katona, Johanna Takács, László Kerner, Zoltán Alföldi, Imre Soós, Tamás Gyömörei, Robert Podstawski, Ferenc Ihász

**Affiliations:** 1Doctoral School of Health Sciences, Faculty of Health Sciences, University of Pécs, 7622 Pécs, Hungary; ifjkernerlaszlo@gmail.com (L.K.); zoltan.alfoldi85@gmail.com (Z.A.); soosimi@gmail.com (I.S.); ihasz.ferenc@ppk.elte.hu (F.I.); 2Physical Education and Sports Centre, Széchenyi István University, 9026 Győr, Hungary; gyomorei@sze.hu; 3Department of Social Sciences, Faculty of Health Sciences, Semmelweis University, 1085 Budapest, Hungary; takacs.johanna@se-etk.hu; 4Department of Tourism, Recreation and Ecology, University of Warmia and Mazury in Olsztyn, 10-719 Olsztyn, Poland; podstawskirobert@gmail.com; 5Doctoral School of Psychology, ELTE Eötvös Loránd University, 1064 Budapest, Hungary

**Keywords:** physical activity, screen time, healthy lifestyle, distance education, coronavirus

## Abstract

Background: High school education took place in the form of distance learning during SARS-CoV-2 pandemic worldwide, including Hungary. Decreased physical activity and an increase in inactive behaviours may lead to an increased risk of obesity, diabetes, and cardiovascular disease. Aim: Our study targeted changes in physical activity (aerobic exercise, muscle strengthening) and screen time in adolescents and young adults during the pandemic. Methods: High school students were interviewed in 66 public schools in 37 Hungarian cities (N = 2508). Survey items on physical activity and screen time were derived from the WHO Health Behaviour of School-aged Children Survey and the Centers for Disease Control Youth Risk Behavior Survey. A 2 × 2 factorial ANCOVA was used to test the effects of gender (male vs. female) and/or age (adolescents vs. young adults) on the reported changes in physical activity and screen time before and during lockdown (covariate: BMI Z-score). Results: The majority of the cohort indicated less physical activity. Aerobic and muscle-strengthening type of exercises significantly decreased, and screen time increased during distance education. Male individuals showed a higher decrease in the level of aerobic exercise, and young adults reported a higher increase in the time spent in front of the screen.

## 1. Introduction

The SARS-CoV-2 coronavirus pandemic (COVID-19) has overall health, social and economic consequences. The pandemic has disrupted normal daily routines worldwide: including attendance teaching among children and adolescents, and across all age groups, depending on the degree of vulnerability [1]. The COVID-19 restrictions have completely changed the daily behaviour of individuals worldwide in 2020, including young people of Hungary.

Under the measures introduced on 26 November 2020, a digital timetable has been introduced in secondary education from grade 9 in Hungary. Educational institutions were instructed to suspend in person class attendance and switch to online e-learning and education. Students were home-bound and digital class attendance, including Physical Education (PE) lessons, was required between 8 a.m.–4 p.m. on school days [2,3].

The global health crisis has resulted in high level inactivity and several studies recognized the negative impact of physical inactivity on health status [4,5]. Ács et al., 2020 [6] pointed out that during the COVID-19 pandemic, decreased levels of physical activity (PA) were detected among young people in higher education, highlighting the need to promote a physically active lifestyle in the face of pandemic-related restrictions [7,8].

As for the daily expected requirement for PA for adolescents (5–17 years)—at least 60 min of moderate-intensity PA has been significantly reduced [9]. The benefits of physical activity, time spent in a sedentary position and adherence to sleep recommendations are independently important components [10,11] of maintaining optimal health [12,13]. The right combinations of PA can be of great importance in maintaining optimal health. Similarly, Saunders et al. [14] have reported that children and young people who comply with 24-h PA rules (high physical activity/high sleep/low sedentary behaviour (SB)) tend to have more favourable adiposity and cardio metabolic indicators compared to those who do not comply with these recommendations. Current evidence suggests that the composition of PA behaviours within a 24-h period can have important implications for health at all ages, and that meeting current 24-h PA guidelines are related to a variety of a range of desirable health indicators in children and young people [15]. However, the restrictions associated with COVID-19 are likely to exacerbate the current, already unfavourable public health situation. Several studies have reported in Canada [16], China [17], Italy [18], Croatia [19] Spain and Brazil [20,21,22]. Ács et al. [6] show that curfew restrictions have led to a significant decrease in all other categories of sporting activity in Hungary [6]. Studies found decreased amounts of PA among university students during the COVID-19 pandemic period [6,23]. 

Understanding the results of studies worldwide, focused on and reported results of the adult population we considered the necessity of a study that assesses PA, screen time (ST) in adolescents and young adults. The aim of this study, therefore, was to conduct a report on changes in PA and ST behaviours during versus before the COVID-19 pandemic lockdown between secondary school students during the digital distance education at the time of COVID-19 lockdown in Hungary. 

We acknowledge that the gender and biological sex of a person are related but not synonymous [24]. In this paper, the terms “girls/boys”and “female/male” refer to biological females and males and their specific health considerations, respectively [25]. Therefore, the information presented here may be useful for biological females and males, as well as individuals of diverse genders [25,26]. 

## 2. Materials and Methods

### 2.1. Participants

Participants were selected from public and private schools as well. Secondary school education takes place in many different types of institutions in Hungary. Age of secondary school students can very between 15–21 years, depending on the form and type of education they choose. Secondary school students (N = 2556) in 66 public education institutions in 37 cities were surveyed from nine regions in Hungary (Figure 1). In total of 48 students from the study sample were excluded; 35 responses were non assessable; 13 students rejected the survey claiming lack of parental agreement. This research was approved by the Ethical Committee of the Medical Research Council (TUKEB), Hungary, under ETK TUKEB ethical permission No. IV/3067- 3/2020/EKU.

### 2.2. Survey

A self-report questionnaire based on the Health Behaviour of School-aged Children (HBSC) [27] and the Youth Risk Behaviour Survey (YRBSS) created by The Centers for Disease Control (CDC) [28] was created. Some questions were modified for during versus before the COVID-19 pandemic lockdown periods. The 15-min-long online survey was located on web page and was accessible for six weeks during distance education while lockdown. Our research team were in touch with school principals and physical education teachers, who encouraged the students to complete the survey. The students completed the survey primarily during online physical education lessons. Information about the aims of the research was given online and parental consent was requested prior to completion. 

Items from CDC’s Youth Risk Behavior Survey [28] were used as the PA measures in this study. Change of vigorous PA was assessed by asking set of question in pairs from respondents such as “On how many days per week did you exercise or participate in physical activity for at least 20 min that made you sweat and breathe hard, such as basketball, soccer, running, swimming laps, fast bicycling, fast dancing, or similar aerobic activities before distance learning period?” and “On how many days per week did you exercise or participate in physical activity for at least 20 min that made you sweat and breathe hard, such as basketball, soccer, running, swimming laps, fast bicycling, fast dancing, or similar aerobic activities during distance learning period?” Similarly, change of muscle-strengthening exercise was measured by asking question pairs “On how many days per week did you do exercises to strengthen or tone your muscles, such as push-ups, sit-ups, or weight lifting before distance learning period?” and “On how many days per week did you do exercises to strengthen or tone your muscles, such as push-ups, sit-ups, or weight lifting during the distance learning period?” At each questions a 0–7-point scale was provided for answers.

### 2.3. Data Analysis

To describe the data, descriptive analysis, shift tables and the distribution of relative frequencies were used. Data were presented as mean ± SD or frequency and proportion. Characteristics of the sample were examined by age groups and gender, using Independent Samples T test, Fisher’s exact test and Pearson’s chi-square test. To examine the effects of gender (females vs. males) and/or age groups (adolescents vs. young adults) on the reported changes in physical activity (measured by the level of aerobic exercise and muscle strengthening) and screen time. A 2 × 2 factorial ANCOVA was used in this model. Dependent variable (reported changes) was calculated as the difference between the number of days/week during and before distance education. Gender (females vs. males) and age groups (adolescents vs. young adults) was used as factors. Finally, we used Body Mass Index (BMI) Z-score as a covariate in this model. BMI is assumed to relate to reported changes (dependent variable) in the physical activity and screen time [29,30], and BMI is also assumed to associate to gender and age [31,32]. The level of significance was set a priori at 0.05. Statistical analysis and visualization were conducted using IBM SPSS Statistics for Windows, Version 25.0 (IBM Corp. Released 2017. Armonk, NY, USA: IBM Corp).

## 3. Results

### 3.1. Characteristics of the Sample

Based on the age, we classified students into two groups, adolescents (A, 56.3% age min-max = 14–17 yr) and young adults (YA, age min-max = 18–21 yr). Age groups showed a non-significant association with gender but revealed a significant association with the BMI category. Thus, BMI Z-score was an appropriate covariate for the analysis. Table 1. shows descriptive sample statistics.

### 3.2. Physical Activity before and during Distance Education

Physical activity was assessed by the self-reported level of aerobic exercise (AE), muscle strengthening (MS), before and during distance education. To ensure unbiased data analysis, inactive students, (e.g., did not do AE or MS before or during distance education were excluded (*n* = 41).

#### 3.2.1. Aerobic Exercise (AE)

Nearly three-quarters of the students showed changes in the level of AE, 1485 students (60.9%) reported a decreased level, and 350 (14.4%) an increased level of AE (Table 2). A total of 55% of students were doing one to three days less AE per week during distance education as before (see Figure 2A).

For studying the gender and age differences of changes in the AE level, students who did not do AE (*n* = 31) or reported no changes (*n* = 601) were excluded. In factorial ANCOVA model gender showed a significant main effect (F(1,1830) = 6.034, *p* = 0.014, η^2^_p_ = 0.003). Male population reported a higher decrease in AE level (M = −1.45, SD = 2.09), than female (M = −1.19, SD = 1.85). Age, gender x age, and the covariate were non-significant.

#### 3.2.2. Muscle Strengthening (MS)

Around 70% of the students showed changes in the level of MS, 1041 students (44.5) reported a decreased level, and 530 (22.6%) an increased level of MS (Table 3.). 35% of students were doing one to two days less MS per week during distance education as before (Figure 2B). 

For studying the gender and age differences of changes in the MS level, students who did not do MS (*n* = 127) or reported no changes (*n* = 769) were excluded. Based on the 2 × 2 ANCOVA model, there was no significant effect of gender and/or age in the changes of the level of MS.

### 3.3. Screen Time (ST) before and during Distance Education

For unbiased analysis, those students who reported ‘zero day’ time spent in front of the screen in the evenings before and during distance education (*n* = 26) were excluded. In the case of the changes in ST, 42.4% of the students showed no changes and more than half of them (54%) reported an increased level of ST. (Table 4). 42.4% of students were in front of the screen in the evenings one to three days more per week during distance education as before (see Figure 2C). 

For studying the gender and age differences of changes in the ST level, students who reported no changes (*n* = 1052) were excluded. In the factorial ANCOVA model age showed a significant main effect (F(1,1425) = 4.280, *p* = 0.039, η^2^_p_ = 0.003). Young adults reported a higher increase in ST level (M = 2.31, SD = 1.76), than adolescents (M = 2.13, SD = 1.79). Age, gender x age, and the covariate were non-significant.

## 4. Discussion

This project is one of the first known studies to examine the early effects of the COVID-19 pandemic on different types of PA and ST among Hungarian high school students. Data were collected during a period of time (November–December 2020) when the most restrictive policies were in place to prevent the spread of the virus, including the closure of primary and secondary schools, the cancellation of team sports and activity classes for youth, and the closure of public parks and playgrounds too. Our findings show a decrease in AE and MS type exercises during the COVID-19 distance education in the samples in comparison with pre-distance education. 

Results from Brazilian and Spanish researchers show that the weekly frequency of 60 min of moderate to high intensity physical activity (MVPA) was significantly reduced in COVID-19 lockdown compared to pre lockdown [20]. In contrast, ST and sleep time duration increased [21]. Very low proportions of participants in Spain (0.3%) and Brazil (7.5%) met the 24-h physical activity requirement [22]. Social constraints, including distance education and forced stay-at-home arrangements, made it difficult for children and adolescents to participate in physical education, sports or community-based organized PA related to school [19]. Another direct consequence of confinement is a decrease in outdoor play, which tends to suggest that this type of physical activity is not part of the daily routine of adolescent children. These results are consistent with other studies that have reported on COVID-19 confinement [33,34]. 

These results showed that the activity pattern of adolescents differed from pre- COVID-19. period. The most commonly reported physical activity in the early COVID-19 period was unstructured free play PA. This pattern was not surprising given school closures and cancellation of team sports/activity classes, with most children spending all day at home and not accessing structured activity activities. However, it contrasts with typical patterns of children’s PA, suggesting that unstructured and free play activities are becoming less common as children’s time is increasingly filled by organised activities. This may also mean that children are not aware of the opportunities offered by their environment and thus look forward to other activities less typical of physical activity during their stay [33]. 

Despite global predictions of a decline in physical activity levels (PAL) among adolescents due to the COVID-19 pandemic and associated isolation and social distancing, empirical evidence is lacking. In the most recent study, Zenic et al., confirmed a larger decline in PAL levels in boys than in girls, which can be explained by the alarmingly low PAL levels in girls. In the run-up to the COVID-19 pandemic, it highlighted the need for further studies that take into account the potential impact of different environmental factors on changes in human life [20]. 

As a whole, these data suggest that in the early COVID-19 period, adolescents spent most of their unstructured leisure time in sedentary activities rather than physical activities. Girls and their older boy peers tended to spend more time in sedentary activities than their younger peers [35]. During the COVID-19 pandemic, this situation may be exacerbated for girls and their older boy peers, as they are at even greater risk of physical inactivity, such as health problems and metabolic dysregulation due to obesity.

The above reasons also provide a strong rationale to study the impact of COVID-19 on the health of adolescents and young adults. Italian researchers have investigated [36] the benefits of regular physical activity, particularly in times of anxiety, crisis and fear. Therefore, it is of concern that in the context of the pandemic, lack of access to regular exercise or exercise routines has led to challenges in immune and physical health, including by leading to the development or exacerbation of existing diseases that have their roots in sedentary lifestyles. Lack of access to exercise and physical activity has also been associated with mental health effects, complex stress or anxiety, which many people experience due to isolation from normal social life. [37] The potential loss of family or friends due to the virus, as well as the impact of the virus on an individual’s economic wellbeing and access to food, exacerbated these effects [38].

The strengths of this study include its large sample size with nationally comparable demographics and the potential of further regional or age-group analysis. There are some limitations of the present study. Its limitations include the fact that all data were collected online and self-reported by participants. Self-reports are subject to recall biases therefore it is possible that they under- or overestimated the time spent in from of the screen or with PA. There are also some the potential response biases of a self-reported questionnaire such as misunderstandings, social desirability and acquiescence bias or extreme responding, etc. At the same time, data collection based on a self-reported questionnaire was the relevant method to answer research questions, even if it has some biases; this method allows the data collection from a large number of study participants with a high response rate. The self-reported measurement of physical activity has some biases. There are a huge number of instruments to assess PA with different format and development. Thus, for assessing PA, we used the Health Behaviour in School-aged Children (HBSC) questionnaire which has been a validated instrument for cross-sectional studies for decades. Finally, it is important to note that we assessed only the frequency of PA. It is recommended for future studies to assess the intensity and time of PA. 

## 5. Conclusions

The prevalence of adolescents and young adults in Hungary samples meeting PA and ST guidelines was low before the COVID-19 pandemic and even worse afterwards during distance education. The effects of a more sedentary lifestyle as a result of distance learning are already being recognised. The health effects of reduced PA and increased ST among young people have further serious negative public health implications. These facts highlight the need to make efforts to support and develop healthy behaviour patterns during a period of distance education in young people.

The authors of the study recommend that during any subsequent quarantine restrictions, the national school nurse service should carry out regular periodic tele-health checks. Promotions of responsible COVID19-related behaviors, follow non-pharmacological interventions such as avoiding crowded indoor settings, wearing a mask in unavoidable indoor social situations, practicing good hand hygiene. Further consultations with physical education teachers and sports coaches could prioritize physical activity in school-age children. The proven positive effects of PA on individuals’ immune system would ultimately help school children, their families, and societies in general reducing COVID infection-spread.

Authorities should prepare detailed plans for adolescents and young adults meeting health-related guidelines during a potential future pandemic and should develop strategies to avoid the potential harmful collateral effects precipitated by pandemic-related restrictions.

## Figures and Tables

**Figure 1 ijerph-18-13024-f001:**
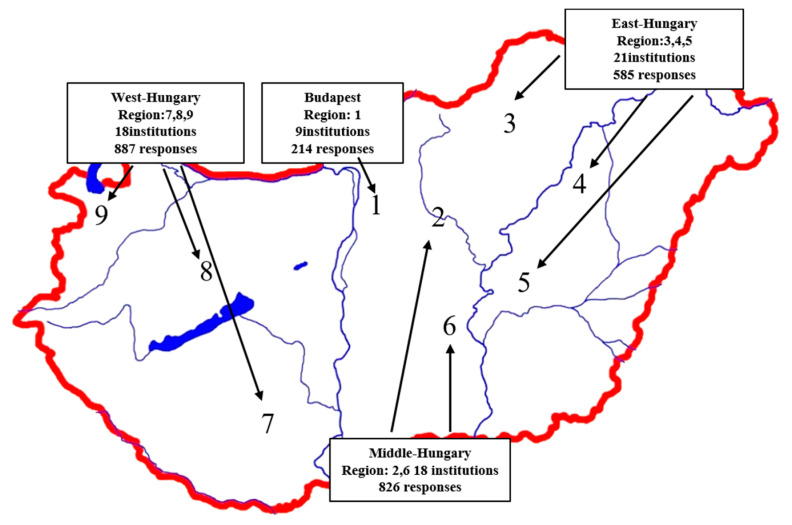
Responses according to geographical locations, institutions and responses.

**Figure 2 ijerph-18-13024-f002:**
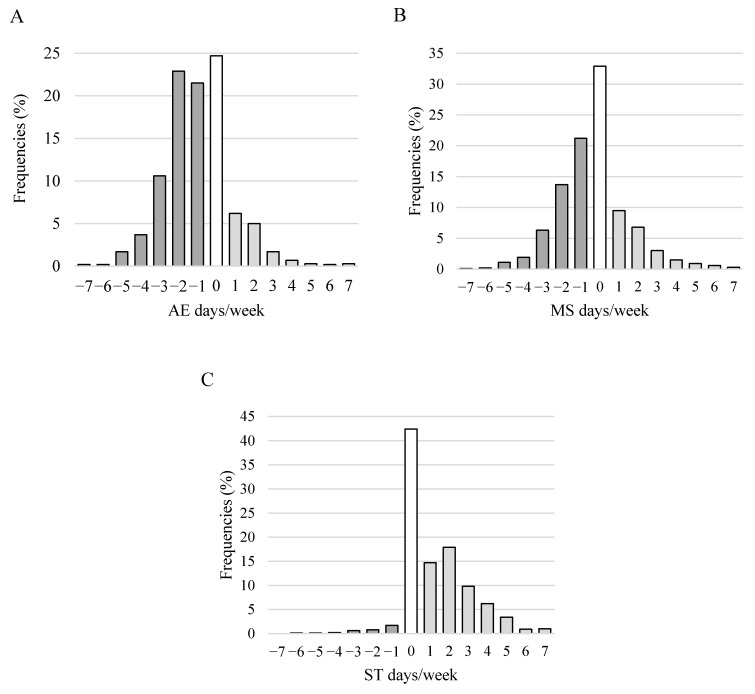
Distribution of the changes (days/week during distance education–before distance education) in the level of aerobic exercise (**A**) and muscle strengthening (**B**), and time spent in front of the screen in the evenings (**C**). Notes. AE: aerobic exercise, MS: muscle strengthening, ST: screen time; AE days/week = AE days/week during distance education–AE days/week before distance education, MS days/week = MS days/week during distance education–MS days/week before distance education, ST days/week = ST days/week during distance education–ST days/week before distance education, white bars: no changes, i.e., the number of the days/week before and during distance education is equal, light grey bars: increase, i.e., days/week during distance education > days/week before distance education, dark grey bars: decrease, i.e., days/week during distance education < days/week before distance education.

**Table 1 ijerph-18-13024-t001:** Characteristics of the sample.

	Study Sample (N = 2508)					
Variables	Adolescents (*n* = 1413)	Young Adults (*n* = 1095)	*p*	Males (*n* = 1072)	Females (*n* = 1436)	*p*
gender, Male *n* (%)	621 (43.9)	451 (41.2)	0.167 ^a^	N/A	N/A	N/A
age (M ± SD)	16.3 ± 0.7	18.6 ± 0.6	N/A	17.3 ± 1.3	17.3 ± 1.3	0.167 ^b^
Weight (kg)	63.3 ± 13.6	66.6 ± 14.5	N/A	71.4 ± 14.7	59.8 ± 11.3	N/A
Height (cm)	171.2 ± 9.3	171.9 ± 9.5	N/A	178.6 ± 7.7	166.2 ± 6.5	N/A
BMI categories ^1^, *n* (%)						
underweight	89 (6.3)	40 (3.7)	<0.001 ^c^	55 (5.1)	74 (5.2)	<0.001 ^c^
normal weight	1146 (81.1)	856 (78.2)		817 (76.2)	1185 (82.5)	
overweigth/obese **	123 (8.7)	131 (11.9)		142 (13.3)	112 (7.8)	
obese	55 (3.9)	68 (6.2)		58 (5.4)	65 (4.5)	

Notes. ^1^ BMI categories base on BMI %ile for 2–19 yr, and on BMI score for ≥20 yr, ** ‘overweight/obese’ terminology based on: Barlow SE and the Expert Committee 2007, ^a^ Fisher’s exact test for association between gender and age-groups, ^b^ Independent Samples T-Test for differences between males and females, ^c^ Pearson’s chi-square test for association between gender/age-groups and BMI categories, N/A statistical analysis is not applicable.

**Table 2 ijerph-18-13024-t002:** Shift table for the level of aerobic exercise before and during distance education.

		AE Days/Week during Distance Education								Total
		0	1	2	3	4	5	6	7	
AE days/week before distance education	0	31 *	6	8	6	5	1	1	7	65
	1	34	**39**	18	22	6	2	1	3	125
	2	59	121	**81**	33	18	12	1	6	331
	3	69	133	157	**143**	38	38	10	8	596
	4	31	85	173	103	**112**	34	24	8	570
	5	31	30	71	146	60	**122**	17	11	488
	6	3	1	13	22	27	38	**50**	6	160
	7	6	3	10	16	12	21	10	**54**	132
Total		264	418	531	491	278	268	114	103	2467

Notes. AE: the level of aerobic exercise means the number of days/week doing aerobic exercise * Do not do AE during and before distance education, bold: no changes in the level of exercise, i.e., AE days/week during distance education = AE days/week before distance education, light grey: increased AE level, i.e., AE days/week during distance education > AE days/week before distance education, dark grey: decreased AE level, i.e., AE days/week during distance education < AE days/week before distance education.

**Table 3 ijerph-18-13024-t003:** Shift table for the level of muscle strengthening (MS) before and during distance education.

		MS Days/Week during Distance Education								Total
		0	1	2	3	4	5	6	7	
MS days/week before distance education	0	127 *	33	24	16	8	10	9	7	234
	1	112	**150**	48	31	13	10	6	6	376
	2	96	175	**209**	64	43	21	9	6	623
	3	64	108	119	**186**	40	33	11	7	568
	4	16	26	59	47	**78**	29	14	10	279
	5	22	14	38	40	27	**84**	7	14	246
	6	3	1	4	12	14	12	**27**	1	74
	7	3	2	3	10	7	4	3	**35**	67
Total		443	509	504	406	230	203	86	86	2467

Notes. MS: the level of muscle strengthening means the number of days/week doing muscle strengthening * not to do MS during and before distance education, bold: no changes in the level of exercise, i.e., MS days/week during distance education = MS days/week before distance education, light grey: increased MS level, i.e., MS days/week during distance education > MS days/week before distance education, dark grey: decreased MS level, i.e., MS days/week during distance education < MS days/week before distance education.

**Table 4 ijerph-18-13024-t004:** Shift table for the level of screen time (ST) before and during distance education.

		ST Days/Week during Distance Education								Total
		0	1	2	3	4	5	6	7	
ST days/week before distance education	0	0 *	8	12	12	13	12	4	25	86
	1	0	**19**	8	25	19	16	9	19	115
	2	0	10	**41**	48	49	58	40	64	310
	3	2	1	8	**64**	71	127	57	85	415
	4	0	0	6	5	**63**	62	72	98	306
	5	0	0	5	3	6	**87**	76	160	337
	6	0	0	3	1	4	7	**56**	91	162
	7	1	2	2	2	8	7	7	**722**	751
Total		3	40	85	160	233	376	321	1264	2482

Notes. ST: screen time means the number of days/week spending in front of the screen in the evenings, * ‘zero day’ time spent in front of the screen in the evenings during and before distance education, bold: no changes in the level of ST, i.e., ST days/week during distance education = ST days/week before distance education, light grey: increased ST level, i.e., ST days/week during distance education > ST days/week before distance education, dark grey: decreased ST level, i.e., ST days/week during distance education < ST days/week before distance education.

## Data Availability

The data presented in this study are available on request from the corresponding author. The data are not publicly available because they belong to minors.

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
