# Peer review of "Physical Activity and Screen Time among Hungarian High School Students during the COVID-19 Pandemic Caused Distance Education Period"

_ijerph, 2021, doi:10.3390/ijerph182413024_

Round 1
Reviewer 1 Report
Strengths:
The research question is of great relevance. The sample size (N=2508 students), which was dependent on response rate for the questionnaire is highly commendable. Overall, the conclusions are congruent with the data and the Research Methods are appropriate.
Suggestions for Improvement:
INTRODUCTION
In the Introduction and the entire manuscript, it would be useful to review and rectify any inconsistencies in grammar, especially with regards to proper tense. e.g. The SARS-CoV-2 coronavirus pandemic (COVID-19) had overall health, social and economic consequences -- since this is an ongoing pandemic, it may be better to say "has had" or "continues to have". There are several other instances where the tense needs to be made consistent with the rest of the article or at least the paragraph/section.
A statement on sex and gender being related but not synonymous terms as included in https://www.mdpi.com/1660-4601/18/19/10271/htm would be useful in the introduction. It may useful to specify that in this paper, the terms males and females or boys and girls refer to biological males or females, respectively. Related to the paper mentioned, above, it would also be appropriate to cite the same paper in the Introduction, since it was published recently in this very journal (IJERPH), and is directly relevant to PA and COVID.
RESULTS
Figure 1 appears pixelated and fuzzy in the compiled PDF. This must be optimized.
Tables and Figures in general. The Table and Figure legends require additional details. It is important that the Tables and Figures along with their legends, can stand alone, and convey all of the essential information without the need for the reader to go back and read parts of the text. Without sufficient information in the legends for the Tables and Figures, the reader may not be able to fully appreciate the main points, and may even misunderstand the data.
DISCUSSION
A note on the limitations of this study must be included - e.g. inaccurate self-reporting
Related to the point above, and related to some of the recommendations made by the authors for prioritizing PA in school-age children, it may be useful to include suggestions regarding periodic telehealth checkups/consultaions that school systems could provide.
Finally, in the interest of promoting public health, it would be useful to please include a note on the need for school children and others to practice responsible COVID-related behaviors -- e.g. getting vaccinated when eligible and continuing to follow nonpharmacological interventions such as avoiding crowded indoor settings, wearing a mask in unavoidable indoor social situations, practicing good hand hygiene, and self-isolating when sick. Reducing COVID infections and spread in society would ultimately help school children, their families, and societies in general be more physically active.
Author Response
Dear Reviewer,
Thank you for your comments and recommendations. Please see below our detailed answers:
INTRODUCTION
In the Introduction and the entire manuscript, it would be useful to review and rectify any inconsistencies in grammar, especially with regards to proper tense. e.g. The SARS-CoV-2 coronavirus pandemic (COVID-19) had overall health, social and economic consequences -- since this is an ongoing pandemic, it may be better to say "has had" or "continues to have". There are several other instances where the tense needs to be made consistent with the rest of the article or at least the paragraph/section.
Response 1: Thank you for your recommendation. An native writer has proofread our paper annd we corrected the grammar accordingly.
A statement on sex and gender being related but not synonymous terms as included in https://www.mdpi.com/1660-4601/18/19/10271/htm would be useful in the introduction.
Response 2: Thank you, we added the references of the recommended article. Ref. No. 24,25,26
It may useful to specify that in this paper, the terms males and females or boys and girls refer to biological males or females, respectively. Related to the paper mentioned, above, it would also be appropriate to cite the same paper in the Introduction, since it was published recently in this very journal (IJERPH), and is directly relevant to PA and COVID.
Response 3: Thanks you, we have inserted a paragraph of statement on sex and gender. Page 2 Lines 75-79. and Reference No. 26
RESULTS
Figure 1 appears pixelated and fuzzy in the compiled PDF. This must be optimized.
Response 4: Thank you, we changed Figure 1 to a better quality picture.
Tables and Figures in general. The Table and Figure legends require additional details. It is important that the Tables and Figures along with their legends, can stand alone, and convey all of the essential information without the need for the reader to go back and read parts of the text. Without sufficient information in the legends for the Tables and Figures, the reader may not be able to fully appreciate the main points, and may even misunderstand the data.
Response 5: Following your advice, we have added more information in the legends for the Tables and Figures. We hope that we gave sufficient information to understand the main points of results in Tables and Figures.
DISCUSSION
A note on the limitations of this study must be included - e.g. inaccurate self-reporting
Response 6: Thank you for your comment. We argee. We inserted a Strengths and Limitations paragraph on Page 8-9 Lines 287-302
Related to the point above, and related to some of the recommendations made by the authors for prioritizing PA in school-age children, it may be useful to include suggestions regarding periodic telehealth checkups/consultaions that school systems could provide.
Response 7: Thanks you, we have inserted a paragraph on page 9, Lines 311- 315.
Finally, in the interest of promoting public health, it would be useful to please include a note on the need for school children and others to practice responsible COVID-related behaviors -- e.g. getting vaccinated when eligible and continuing to follow nonpharmacological interventions such as avoiding crowded indoor settings, wearing a mask in unavoidable indoor social situations, practicing good hand hygiene, and self-isolating when sick. Reducing COVID infections and spread in society would ultimately help school children, their families, and societies in general be more physically active.
Response 8: Thanks you, we have inserted a paragraph on page 9, Lines 315- 319.

Reviewer 2 Report
Please see attached document for comments and suggestions.

Author Response
Dear REVIEWER,
Thank you for taking time of reviewing our paper. Please find below our responses.
Overall This study examined the differences in aerobic exercise, muscular strengthening, and screen time in adolescent and young adult students. The findings suggest a decrease in aerobic exercise and muscular strengthening activities during the COVID-19 distance education in comparison with pre-distance education. This study has potential for publication, but revisions are needed before the final decision can be made. The English is a bit broken and not completely correct throughout the paper. It is advised to allow a native English-writing individual to proofread the manuscript.
Response 1: Thank you for pointing out that. We have asked mative writers proofreading help and corrected the paper according to his recommendations.
**How is this different than the Acs, 2020 study?
Response 2: Acs at al. research targeted adult population only. Our study focused on secondary school students, who are in the age category between 15-19.
The survey used, called Health Behaviour of School-aged Children (HBSC) and the Youth Risk Behaviour Survey (YRBBS) are for youth, it seems. The study sample included those above the youth age, since the ages were 15-21.
Response 3: HBSC questioner contains questions about PA that is not only relevant to 15 years olds but for older secondary school students as well. We do find these questions applicable for teenagers and young adults. Hungarian secondary school system is diverse and manifold. Students normally start at age 15 and complete their studies between age 18-21. It depends on the type of education they choose. E.g. secondary grammar school education takes 4 or 5 years. Vocational training can be 3 or 4 years, but sometimes students stay in the vocational education system for an other 1 or 2 years. Some students do not start primary education at age 6 but age 7 therefore their age at the time of graduation can be 20 or 21 years old.
Additionally, the HBSC link in your references did not work.
Response 4: Thank you. We corrected the link, that is: Health Behaviour in School-aged Children: h https://www.uib.no/en/hbscdata/113290/open-access Available at https://filer.uib.no/psyfa/HEMIL-senteret/HBSC/2006_Mandatory_Questionnaire.pdf
The YRBBS however, does assess youth and young adults, which is the survey you used to assess PA measures. Why was the HBSC used then?
Response 5: PA related questions were used from the questionnaire such as on page 9. “Physical activity is any activity that increases your heart rate and makes you get out of breath some of the time. Physical activity can be done in sports, school activities, playing with friends, or walking to school. Some examples of physical activity are running, brisk walking, rollerblading, biking, dancing, skateboarding, swimming, soccer, basketball, football, & surfing.”
Abstract
Sedentary behavior was measured as screen time only. Change this to screen time as screen time, although correlated with sedentary behavior, is not necessarily sedentary behavior.
Response 6: We do agree with your point. We acknowledge that SB is covering much broader mound while ST is a kind of SB but smaller set. For the sake of being more precise we have changed SB to ST in our study.
Introduction
Page 1, Line 37: What do you mean by “attendance teaching”? Please clarify.
Response 7: Thank you. We agree that “attendance teaching” may not be the best expression. We wanted to refer to a normal pre-pandemic time when students learnt in classrooms together at school. To be more precise we change the expression to “in person teaching”.
Page 2, Line 46-47: Please add citations for this first sentence in the paragraph.
Response 8: Thank you for pointing out that, we have added references to the sentence. > [4,5].
Page 2, Line 50: add a citation for high school students since your study measured individuals 15-21 years old and the study you cited is just university students.
Response 9: Thank you for your suggestion. We added citations to that sentence. > [7,8].
Page 2, Line 71: Physical activity has already been defined and abbreviated but you define and re-abbreviate it again.
Response 10: Thank you for pointing out that, we have corrected it.
Survey
What questions were modified for during versus before the COVID-19 pandemic lockdown periods?
Response 11: PA related questions were assessed by asking question pairs form respondents such as “On how many days per week did you exercise or participate in physical activity for at least 20 minutes that made you sweat and breathe hard, such as basketball, soccer, running, swimming laps, fast bicycling, fast dancing, or similar aerobic activities before distance learning period?” and “On how many days per week did you exercise or participate in physical activity for at least 20 minutes that made you sweat and breathe hard, such as basketball, soccer, running, swimming laps, fast bicycling, fast dancing, or similar aerobic activities during distance learning period?” Similarly, change of muscle-strengthening exercise was measured by asking question pairs “On how many days per week did you do exercises to strengthen or tone your muscles, such as push-ups, sit-ups, or weight lifting before distance learning period?” and “On how many days per week did you do exercises to strengthen or tone your muscles, such as push-ups, sit-ups, or weight lifting during the distance learning period?” The same way we gave question pairs to assess ST before and after lock down.
The process of participants taking the survey during versus before the pandemic lockdown periods needs to be described in better detail. For example, when was test was taken before the lockdown and when was it taken during the lockdown? Or was the survey taken that assessed both before and after the lockdown at the same time?
Response 12: The survey was taken during the lock down period when students were home-bound and digital education was in place. Responses given were both before and after the lockdown at the same time. We understand the bias of the subjectivity of responses as students remembered back to the time of before lock down when filled in the questionnarie during lock down.
Response 13: Thank you for your suggestion. We argee. We inserted a strengths and limitations paragraph on Page 8-9 Lines 287-302. The strengths of this study include its large sample size with nationally comparable demographics and the potential of further regional or age-group analysis. Its limitations include the fact that all data were collected online and self-reported by participants. Self-reports are subject to recall biases therefore it is possible that they under- or overestimated the time spent in from of the screen or with PA.
E.g. considering that all the information was self-reported by participants, it is possible that they under- or overestimated the time spent in from of the screen or with PA.
Data analysis The ANCOVA needs to be described better. Explain that the ANCOVA compared pre versus during the lockdown periods. Describe and provide citations for why BMI was used as a covariate in the models.
Response 14: We thank the reviewer for this comment. Thus, we have described data analysis ANCOVA in more detailed in the data analysis section. “To examine the effects of gender (females vs. males) and/or age groups (adolectends vs. young adults) on the reported changes in physical activity (measured by the level of aerobic exercise and muscle strengthening) and screen time, we used a 2x2 factorial ANCOVA. In this model, dependent variable (reported changes) was calculated as the difference between the number of days/week during and before distance education”. Viz., we subtracted the number of days/week before distance education from the number of days/week during distance education. It was an important step for evaluating the reported changes in physical activity and screen time, because if we use the mean of the days/week before and after in a repeated measure, we could not have caught the real changes.
We have described and provided citations for why BMI was used as a covariate in the models.
“Gender (females vs. males) and age groups (adolescents vs. young adults) was used as factors. Finally, we used BMI Z-score as a covariate in this model. BMI is assumed to relate to reported changes (dependent variable) in the physical activity and screen time [29,30], and BMI is also assumed to associate to gender and age [31, 32]”.
Furthermore, we also found a significant association between gender and BMI in the study sample.
Results 2508 students participated in the study. How many were contacted in total?
Response 15: We collected data from 66 secondary schools. We received 2543 Responses from students but 35 were non assessable. Further 13 students rejected the survey claiming that they did not receive parental agreement.
Were there any incomplete surveys? Include this information.
Response 16: Thank you for your point. We inserted the data on Page 2. lines 88-89. A total of 2543 students participated in the present study (47.2% male). Unfortunately, 35 Responses were non assessable. Further 13 students rejected the survey claiming that they did not receive parental agreement.
What were the age ranges for adolescents and young adults?
- Table 1: Decimals need to replace the commas in the data.
- Table 1: The first line of the “Notes” is clear.
Response 17: Thank you for this comment. We have corrected decimals in Table 1. We have also added age ranges in the text of the Characteristics of the sample. Based on the previous comments, we have added a more detailed legend to Table 1.
- “Based on the age, we classified students into two groups, adolescents (A, 56.3%; age min-max = 14-17 yr) and young adults (YA, age min-max = 18-21 yr).” A total of 2508 students participated in the present study (42.7% male).
Page 4, Lines 137-138: This sentence is not clear. Are these inactive students in regards to school or physical activity? Please clear this up.
Response 18: Thank you for this comment. We have rewritten this sentence.
“To ensure unbiased data analysis, we excluded students who did aerobic exercise and muscle strengthening neither before nor during distance education; the number of the reported days/week was zero in both before and during distance education (n=41).”
Tables 2-4: The * is not clear as what “not to do __(variable)” is. Please make this more clear by saying something such as, “no ST pre or during”. Provide the statistical analyses data (e.g., p-values) in the text in the results. Figures A and B: These figures are not clear. This reviewer suggests generating bar graphs to illustrate the differences in AE days/week pre to during distance education and MS days/week pre to during distance education. The questionnaires used stated that it assessed how much time PA is engaged in each day. Where is the data from the participants on this measure?
Response 19: Thank you for this comment. We have added detailed description in the legends for the Tables and Figures. We hope that we gave sufficient information to understand the main points of results in Tables and Figures.
“Provide the statistical analyses data (e.g., p-values) in the text in the results.”
Shift tables and the distribution of the changes graphed in Figure 2 are descriptive statistics, thus p-values are irrelevant.
“This reviewer suggests generating bar graphs to illustrate the differences in AE days/week pre to during distance education and MS days/week pre to during distance education. The questionnaires used stated that it assessed how much time PA is engaged in each day. Where is the data from the participants on this measure?”
Response 20: Thank you for this suggestion. Regarding Figure 2, we graphed the difference in AE/MS/ST days/week before to during distance education. We subtracted the number of days/week before distance education from the number of days/week during distance education. It was an important step for evaluating the reported changes in physical activity and screen time, because if we use the mean of the days/week before and after in a repeated measure, we could not have caught the real changes. Based on these condiserations, we believe that the distribution of the changes graphed in Figure 2 add more information to and ensure better understandings in the results in the shift tables. It is important to note that we assessed PA and screen time cross-sectionally with two questions that related to „On how many days per week did you do...before?" and "On how many days per week did you do...during? Thus, we assessed the frequency per week (days/week) and not time on a day. We have also added this as a limitations of our study in the new Limiations section.
Discussion
Page 7, Lines 206-209: Add citations after “lockdown” and “increased”.
Response 21: Thank you for your suggestion. We added citations after lockdown [20] and increased [21]. Now these sentences are on page 8 lines 243-244.
Page 8, Lines 244-246: Add citations after “psychology” and “men”.
Response 22: Thank you for your recommendation. After a small consultation with the co-authors we came to an agreement that we delete this parapgraph, used to be Lines 243-249, because we feel that the paragraph does not fit into the train of thoughts.
Page 8, Lines 243-249:
This paragraph is unclear and seems to jump to conclusions. Clear up and perhaps expand on the evidence that suggests that a decline in PA levels is most likely to affect the mental well-being of women.
Response 23: Thank you. After a small consultation with the co-authors we came to an agreement that we delete this parapgraph, used to be Lines 243-249, because we feel that the paragraph does not fit into the train of thoughts.
